# Multi-Fault Classification and Diagnosis of Rolling Bearing Based on Improved Convolution Neural Network

**DOI:** 10.3390/e25050737

**Published:** 2023-04-29

**Authors:** Xiong Zhang, Jialu Li, Wenbo Wu, Fan Dong, Shuting Wan

**Affiliations:** 1Hebei Key Laboratory of Electric Machinery Health Maintenance & Failure Prevention, Baoding 071003, China; hdjxzx@ncepu.edu.cn; 2Department of Mechanical Engineering, North China Electric Power University, Baoding 071003, China

**Keywords:** convolution neural network, rolling bearing, multi-classification problem

## Abstract

At present, the fault diagnosis methods for rolling bearings are all based on research with fewer fault categories, without considering the problem of multiple faults. In practical applications, the coexistence of multiple operating conditions and faults can lead to an increase in classification difficulty and a decrease in diagnostic accuracy. To solve this problem, a fault diagnosis method based on an improved convolution neural network is proposed. The convolution neural network adopts a simple structure of three-layer convolution. The average pooling layer is used to replace the common maximum pooling layer, and the global average pooling layer is used to replace the full connection layer. The BN layer is used to optimize the model. The collected multi-class signals are used as the input of the model, and the improved convolution neural network is used for fault identification and classification of the input signals. The experimental data of XJTU-SY and Paderborn University show that the method proposed in this paper has a good effect on the multi-classification of bearing faults.

## 1. Introduction

As the core component of rotating machinery, the health of a rolling bearing will have a direct impact on the normal operation of mechanical equipment [1,2,3]. When the bearing is partially damaged or defective, it may cause noise and vibration abnormalities at the light level, or damage to the equipment at the heavy level [4,5]. Therefore, timely and effective fault identification and diagnosis of a rolling bearing is of great significance.

In the past, most of the rolling bearing fault studies were based on traditional methods, such as the use of variational modal decomposition, empirical modal decomposition, and other algorithms to decompose the bearing vibration signal, and then select its components for analysis [6,7]. However, traditional methods rely more on expert experience, and too much manual intervention will inevitably have a certain impact on the diagnosis results [8]. Now, with the improvement of intelligence, the collection rate and order of magnitude of various bearing data have been widely improved, which also lays a good foundation for the deep learning method to enter the field of bearing fault diagnosis. As an intelligent research method commonly used at present, deep learning has a strong adaptive extraction ability, which effectively reduces manual intervention and empirical error in the process of bearing data analysis [9]. Therefore, it is gradually applied to the field of bearing fault diagnosis by more and more people. Hoang [10] used a CNN model to directly analyze gray vibration images, and achieved good results in a noise environment. Che [11] extracted time-domain features from the original vibration signal, converted these features into grayscale images, and combined with time series to build multimodal samples. CNN and DBN networks are used to process gray image and time series samples, respectively, and, finally, mode fusion is carried out. Compared with single model analysis, higher fault diagnosis accuracy is achieved. Zou [12] combined a discrete wavelet transform and improved a deep trust network to convert vibration signal into two-dimensional time-frequency map, and recognize the correlation between fault characteristics and fault types through improved DBN after preprocessing the time-frequency map, so as to identify and diagnose the fault status of traction motor bearings. He [13] optimized the parameters of the variational mode decomposition, and then use the optimized VMD algorithm to decompose the original signal into a series of intrinsic mode components, and calculate the energy entropy of each component to build the eigenvector. Finally, the eigenvector is input into the inverse residual convolution neural network model, and the bearing fault is effectively diagnosed. Li [14] studied the multi-source domain learning method and combined it with unsupervised integrated learning, which has achieved excellent results in solving the single source single target problem in transfer learning. Sinitsin [15] used CNN and MLP models together to process different types of input data at the same time, successfully detected and located bearing defects, and achieved high accuracy. Chen [16] used two convolutional neural networks with different kernel sizes to extract signal features of different frequencies from the data, and then used the LSTM algorithm to identify fault types, achieving high average accuracy and demonstrating its excellent performance in noisy environments. Kumar [17] combined an adaptive gradient optimizer with a deep convolutional neural network and used multiple MEMS accelerometers to collect vibration data for analysis, which was used for early fault detection in SCIM and achieved good results. Huang [18] adds multi-scale convolutional layers to traditional convolutional neural networks to enhance the discriminative ability of signals under different fault conditions by integrating multi-scale information of the original signal. Additionally, it achieved higher classification accuracy than the original CNN in both normal and noisy environments.

The above studies are based on the analysis of the situation that the classification of bearing data is less. However, under some complex conditions, there are many kinds of bearing faults coexisting. It is difficult to achieve the expected effect by using ordinary models to identify and diagnose this. In order to solve this problem, this paper proposes a convolution neural network for bearing multiple fault classification (MFCNN) under multiple working conditions. This method improves the traditional convolution neural network and adopts a simple structure of three-layer convolution to simplify the model. The average pooling layer is used to replace the common maximum pooling layer, the global average pooling layer is used to replace the common full connection layer, and the batch normalization layer is used to optimize the model. Moreover, multiple fault signals under multiple working conditions are used as the input of the model for identification and diagnosis. Comparing this method with other diagnosis methods, the results show that the method proposed in this paper has faster operation speed and higher accuracy. Therefore, the method proposed in this paper has important significance for multi-fault diagnosis of rolling bearings and has a more reliable engineering application value.

This paper is summarized as follows. Section 2 introduces the basic theoretical knowledge of the proposed method. Section 3 describes the structure and specific parameter settings of the model. Section 4 describes the specific process of bearing diagnosis. Section 5 introduces the process and result analysis of using XJTU-SY data for experiments. Section 6 introduces the process and result analysis of conducting experiments using Paderborn University bearing data. Section 7 summarizes and evaluates the research methods.

## 2. Basic Theory

### 2.1. Convolution Neural Network

CNN is a feedforward neural network with strong feature extraction capability [19]. At the same time, it can effectively reduce the complexity and computation of the model through local weight sharing. The convolution process replaces the complicated feature extraction process in traditional machine learning, and realizes intelligent fault diagnosis. The traditional CNN structure is composed of an input layer, convolution layer, pooling layer, full connection layer, and output layer. The convolution layer and pooling layer are usually several and are connected alternately [20,21,22]. The model is shown in Figure 1. The MFCNN proposed in this paper is an improved model based on traditional convolution neural network.

The function of the input layer is to receive the signals from the neural network. In this paper, the bearing multi-fault data are used as the signal input.

The function of convolution layer is to perform convolution operation on the input signal to extract important features [23], and its expression is as follows:(1)$$x_{i+1}=w_{i}⊗x_{i}+b_{i}$$
where $x_{i}$ is the current input characteristic matrix, $x_{i+1}$ is the calculated characteristic matrix, $w_{i}$ is the convolution kernel weight parameter, $b_{i}$ is the offset parameter, and ⊗ is the convolution operation.

The pooling layer is generally used to reduce the dimension of features extracted from convolution layer [24]. There are two common pooling operations in convolutional neural networks, namely, maximum pooling and average pooling. The function of maximum pooling is to extract the maximum value of all data in the pooling window, and the average pooling is used to calculate the average value of all data in the pooling window. The schematic diagram of the two operations is shown in Figure 2, in which the size of the pooling window is 2 × 2 and the step size is 2. The maximum pooling operation takes the maximum value in each of the four small areas to form a new matrix. The average pooling operation takes the average of the values in each small area to form a new matrix. After the two pooling operations, the dimensions of the original matrix have been reduced, but the different operation modes of the two methods also determine their respective advantages. In image processing, the maximum pooling operation can make the feature information more sensitive to texture and contour information, which is beneficial to better identify the key targets in the image. The average pooling operation can make the feature information more sensitive to the background information, but it is easy to blur the image. The model in this paper directly analyzes the bearing fault data. Considering the difference between the values and pictures, the effect of pooling layer is explored in the experiment, and the effect of maximum pooling and average pooling on the accuracy of calculation is compared. We used the maximum pooling layer and the average pooling layer in the MFCNN model, respectively, to conduct five experiments and record the test accuracy. The experimental results are shown in Figure 3. According to the experimental results, the accuracy of using average pooling is generally higher than that of using maximum pooling. Therefore, in this model, the average pooling layer is used to replace the maximum pooling layer commonly used in image processing.

The full connection layer is responsible for transforming the two-dimensional feature matrix output after the previous series of processing into a one-dimensional vector, integrating features together, and greatly reducing the impact of feature location on classification. However, too many parameters of the full connection layer will increase the difficulty of network training. This paper uses the global average pooling layer to replace the full connection layer, which can not only realize the function of the full connection layer, but also reduce the number of parameters and avoid over-fitting [25]. Compared with the full connection layer, using the global average pooling technology can make the model structure simpler, thus speeding up the calculation speed.

The output layer is located at the end of the whole neural network structure, and classifies and outputs the features obtained from the front.

### 2.2. Batch Normalization

Batch normalization is a data normalization method. The operation process of batch normalization of data from any output in training is shown below.

Calculate the mean value of batch processing data:(2)$$\mu =\frac{1}{m}\overset{i=1}{\underset{m}{\sum^{\;}}}x_{i}$$

Calculate the variance of batch processing data:(3)$$\sigma^{2}=\frac{1}{m}\overset{i=1}{\underset{m}{\sum^{\;}}}\left(x_{i}-\mu \right)$$

Normalize data:(4)$${\hat{x}}_{i}=\frac{x_{i}-\mu }{\sqrt{\sigma^{2}+\varepsilon }}$$

Scale transformation and offset:(5)$${\hat{y}}_{i}=\gamma x_{i}+\beta$$
where m represents the size of the batch; ε is a constant term to ensure numerical stability; γ and β are scale factors and translation factors, respectively, which can be learned through the network; and ${\hat{y}}_{i}$ is the output of BN layer. Through batch normalization operations, the output data of each layer can always present a normal distribution, which greatly improves the training efficiency of the model [26].

## 3. MFCNN Model

The MFCNN method proposed in this paper is an improvement on the traditional convolution neural network. It combines the advantages of traditional CNN and increases the ability of multi-fault recognition. The bearing multi-fault signal is taken as the input of the model, and the shallow neural network with three convolution layers is used for analysis to reduce the burden of calculation. The neural network model uses the average pooling layer instead of the common maximum pooling layer, which can improve the classification accuracy. It reduces the number of parameters of the model by replacing the full connection layer with the global average pooling layer, so as to simplify the model and reduce the calculation pressure. In addition, BN layer is added to the neural network after convolution. The existence of BN layer can accelerate the speed of training and convergence and prevent over-fitting. The specific model parameters are shown in Table 1. After many comparisons in the experiment, it is determined that MFCNN method can greatly improve the accuracy of the diagnosis results when bearing multi-fault classification, and greatly improve the diagnosis effect of traditional neural network.

## 4. Fault Diagnosis Process

The fault diagnosis process is shown in Figure 4, which is mainly divided into four parts. The first part is the selection of original signals. In this paper, XJTU-SY bearing data and QPZZ-II bearing data are used as the original signals, respectively, and several kinds of fault data are selected to build the input information of the model. The second part is data preprocessing, which scrambles and reorganizes the selected data, and divides them into training sets and test sets according to the ratio of 7:3. The third part is model training and parameter adjustment. Through multiple training and analysis, the best parameters are gradually determined. The fourth part is the identification and diagnosis of bearing faults, and the visual analysis of the diagnosis results.

## 5. Experimental Data Analysis of XJTU-SY Bearing

### 5.1. Data Preprocessing

The data used in this experiment is from the joint laboratory of mechanical equipment health monitoring established by Professor Lei Yaguo of Xi’an Jiaotong University and Zhejiang Changxing Sumyoung Technology Co., Ltd. (Huzhou, China). The signal collected in the experiment is the time domain vibration signal of the bearing. The experimental platform of these data is composed of AC motor, motor speed controller, rotating shaft, support bearing, hydraulic loading system, and test bearing, as shown in Figure 5. The rolling bearing model used is LDK UER204, and the specific specifications are shown in Table 2. Several common bearing conditions are shown in Figure 6.

The experimental data include three types of working conditions. In condition 1, the bearing speed is 2100 r/min and the radial force is 12 KN. In condition 2, the bearing speed is 2250 r/min and the radial force is 11 KN. In condition 3, the bearing speed is 2400 r/min and the radial force is 10 KN.

In this experiment, the inner ring fault, outer ring fault, cage fault, mixed fault data, and health status data under three working conditions are selected for dataset construction, and the label settings are shown in Table 3. The 120,000 sampling points are taken for each state and divided into 300 groups with 400 points in each group. The 300 groups of data are divided into training group and test group, of which 210 groups are put into training, and the remaining 90 groups are tested.

Select a sample from various types to draw time-domain and frequency-domain diagrams, as shown in Figure 7 and Figure 8. As shown in the figure, it is difficult to diagnose fault types solely through time-domain and frequency-domain diagrams, and a large amount of manpower is required, making it difficult. Therefore, it is necessary to introduce a convolutional neural network model for recognition.

### 5.2. Experiment and Result Analysis

The deep learning framework used in the experiment is Tensorflow, and the computer configuration is Core (TM) i5-8265U CPU processor and NVIDIA GeForce MX230 graphics card.

Input the data into the MFCNN model for training, and the training stops after 200 iterations. This experimental model uses the Adam optimizer to automatically optimize the learning rate, making the results more accurate. The cross-entropy loss function is used as the objective function to guide the learning of network parameters. The accuracy curve of training and testing is shown in Figure 9, and the loss curve is shown in Figure 10.

It can be seen from Figure 9 and Figure 10 that the accuracy curve of the training set has completely converged after about 25 iterations, and the accuracy rate has reached 100%. The loss curve decreases rapidly with the iteration, converges completely at about 50 times, and the loss is infinitely close to zero. The accuracy curve of the test set converges completely after about 100 iterations, reaching 99.66%. The loss curve decreases rapidly with the iteration, converges completely at about 100 times, and is infinitely close to zero. Figure 11 shows the confusion matrix of the test set. Its abscissa is the forecast label, and its ordinate is the actual label. It can be seen from the confusion matrix that in the test process, the recognition accuracy of other categories has reached 100%, except for some slight errors on the categories with labels 8 and 10. Figure 12 is a visual diagram of the overall process during training. From the diagram, it can be seen that the distribution of the original data is relatively scattered, with various data mixed together. As the training process progresses, different types of data points gradually disperse, while data points of the same type gradually gather and finally completely separate, achieving excellent classification results. This indicates that the classification effect after training is better. At the same time, it has also been proven that the method proposed in this paper has good diagnostic performance for the multi-fault classification problem of rolling bearings.

### 5.3. Comparison of Different Fault Diagnosis Methods

In order to verify the superiority of the proposed method, it is compared with three typical diagnostic methods. During training, the batchsize is set to 128 and the number of iterations is 500. Visualize the final test curve, as shown in Figure 13 and Figure 14. Count the number of parameters in various methods, as shown in Table 4. It can be seen from the results that the method proposed in this paper has the best effect, the accuracy curve and the loss curve converge the fastest in all methods, the accuracy rate reaches the highest 99.83% in all methods, and the loss is the lowest in all methods, which is infinitely close to zero. The number of parameters in this model is the least among various methods, only 28429. The fewer parameters make the computer run with less burden and faster operation speed. Models such as ShuffleNetV1, GhostNet, and MobileNetV2 have a large number of parameters, resulting in longer training times. The test curves of the three typical models fluctuate greatly and are difficult to converge effectively. Compared with the model proposed paper, its accuracy is lower, the loss is greater, and the effect is not satisfactory.

## 6. Analysis of Experimental Data on Paderborn University Bearings

The bearing data used in this experiment are real bearing damage sample data generated by Paderborn University through accelerated life testing. The experimental bearing is 6203 Deep Groove Ball Bearing. The bearing test bench is shown in Figure 15, and the bearing failure situation is shown in Figure 16. The Paderborn University bearing experiment divided the bearing damage situation into five levels, with 1 to 5 indicating that the damage is becoming increasingly severe. In this paper, time domain vibration signals from 13 samples were selected for analysis in the experiment. The specific situation and label settings of the samples are shown in Table 5.

Select a sample from various types to draw time-domain and frequency-domain diagrams, as shown in Figure 17 and Figure 18. As shown in the figure, similarly, it is difficult to diagnose fault types solely through time-domain and frequency-domain diagrams. Therefore, it is necessary to introduce a convolutional neural network model for recognition.

Similarly, divide the data in each state into 300 groups with 400 sampling points per group. The 300 sets of data are divided into training and testing groups, with 210 groups trained and the remaining 90 groups tested.

The data are input into the MFCNN model for identification and diagnosis, and the training and test accuracy curve is shown in Figure 19, and the training and test loss curve is shown in Figure 20.

It can be seen from Figure 19 and Figure 20 that the accuracy curve of the training set has fully converged after about 50 iterations, and the accuracy rate has reached 100%. The loss curve decreases rapidly with the iteration, converges completely at about 100 times, and the loss is infinitely close to zero. The accuracy rate of the test set converges completely after about 150 iterations, reaching 85.38%. The loss decreases rapidly with the iteration and fully converges at approximately 150 iterations. Figure 21 shows the confusion matrix of the test set. Its abscissa is the forecast label, and its ordinate is the actual label. From the confusion matrix, it can be seen that, in the process of testing, except for some errors in some categories, the recognition accuracy of most categories has reached more than 80%. Figure 22 is a visual diagram of the overall process during training. From the diagram, it can be seen that the distribution of the original data is relatively scattered, with various data mixed together. As the training process progresses, different types of data points gradually disperse, while data points of the same type gradually gather and finally completely separate, achieving excellent classification results. This indicates that the classification effect after training is better. At the same time, it has once again been proven that the method proposed in this paper has good diagnostic performance for the multi-fault classification problem of rolling bearings.

Figure 23 and Figure 24 show the comparison of the test results of the four methods. It can be seen from the two figures that the MFCNN method in this paper has significantly better test results than other methods under the same batch size and iteration times.

## 7. Conclusions

In order to solve the problem that the bearing diagnosis becomes more difficult under the condition of multiple working conditions and faults, this paper proposes an MFCNN method. The main advantages of this method are as follows. (1) The shallow neural network structure of three-layer convolution is adopted to solve the problem and reduce the burden of hardware in the calculation process; (2) the average pooling layer is used to replace the common maximum pooling layer, which significantly improves the diagnostic accuracy; (3) replacing the flattening layer and the full connection layer with the global average pooling layer greatly reduces the number of neurons in the model and prevents over-fitting; and (4) the BN layer is used to optimize the neural network model, which accelerates the speed of training and convergence, and enhances the stability of the model. The bearing multi-fault data under various working conditions are input into the convolution neural network for fault identification and diagnosis, and excellent results are obtained, which proves the effectiveness and superiority of the proposed method in dealing with bearing multi-fault classification problems. At the same time, the method proposed in this paper also has some shortcomings. As shown in the paper, for some data that are difficult to classify, such as Paderborn bearing data, although the diagnostic performance is much better than some typical lightweight algorithms, the accuracy is still slightly poor. Therefore, continuing to improve the classification performance of the model for difficult to classify data is also a direction for future research and optimization.

## Figures and Tables

**Figure 1 entropy-25-00737-f001:**
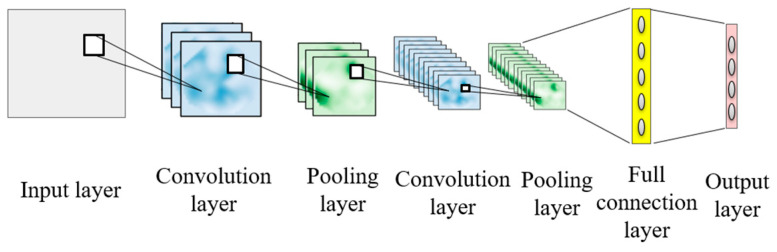
Traditional CNN structure model.

**Figure 2 entropy-25-00737-f002:**
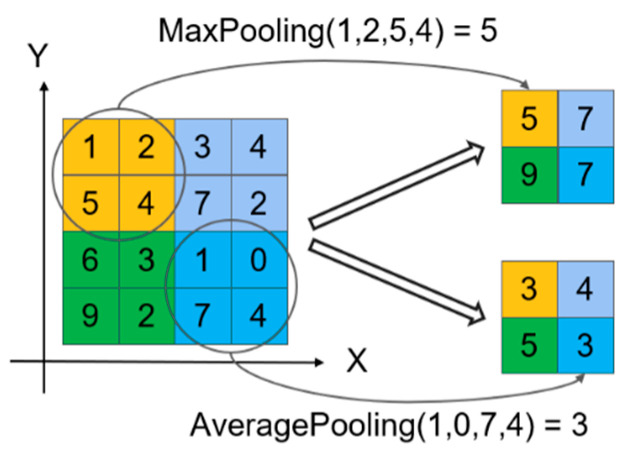
Pooling diagram.

**Figure 3 entropy-25-00737-f003:**
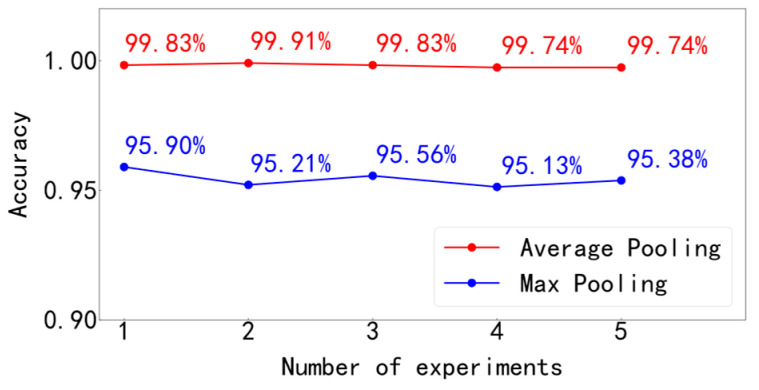
Comparison of different pooling effects.

**Figure 4 entropy-25-00737-f004:**
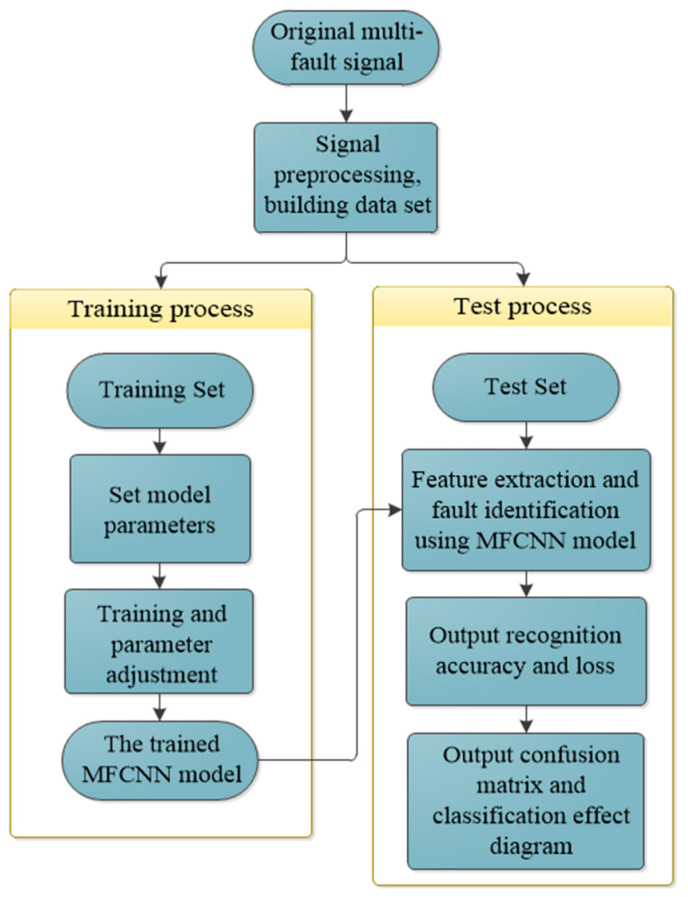
Fault diagnosis process.

**Figure 5 entropy-25-00737-f005:**
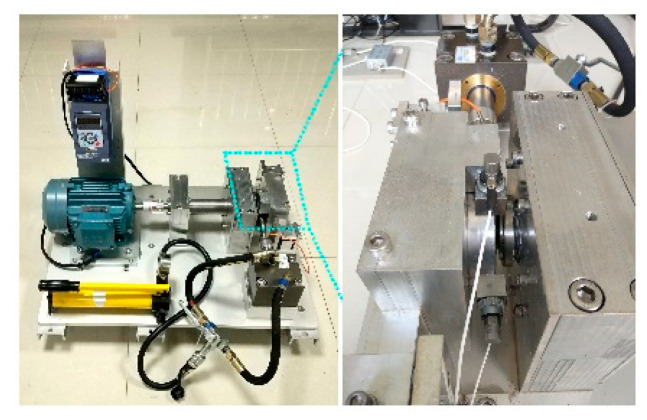
XJTU-SY bearing test bench.

**Figure 6 entropy-25-00737-f006:**
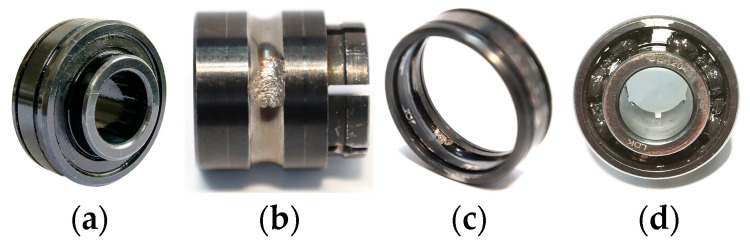
Schematic diagram of various conditions of bearings. (**a**) Normal conditions. (**b**) Inner ring fault. (**c**) Outer ring fault. (**d**) Cage fault.

**Figure 7 entropy-25-00737-f007:**
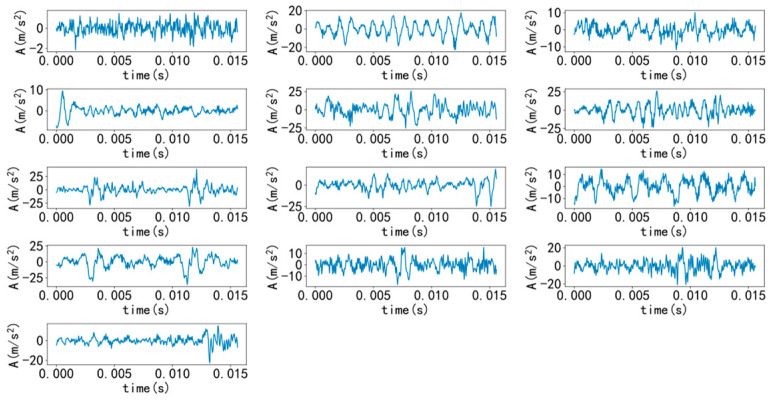
Various types of time domain diagrams.

**Figure 8 entropy-25-00737-f008:**
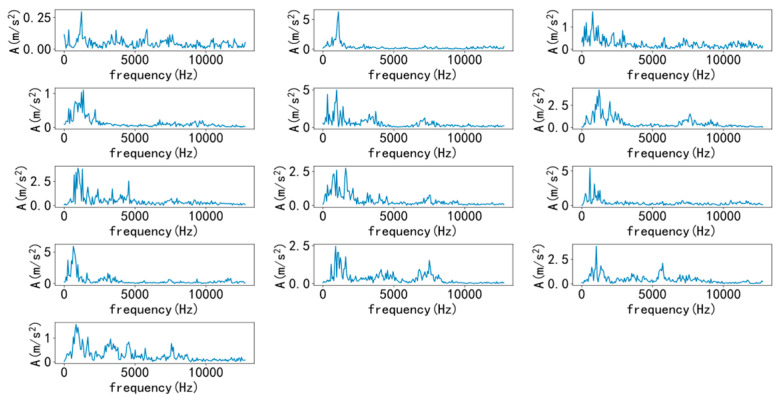
Various types of frequency domain diagrams.

**Figure 9 entropy-25-00737-f009:**
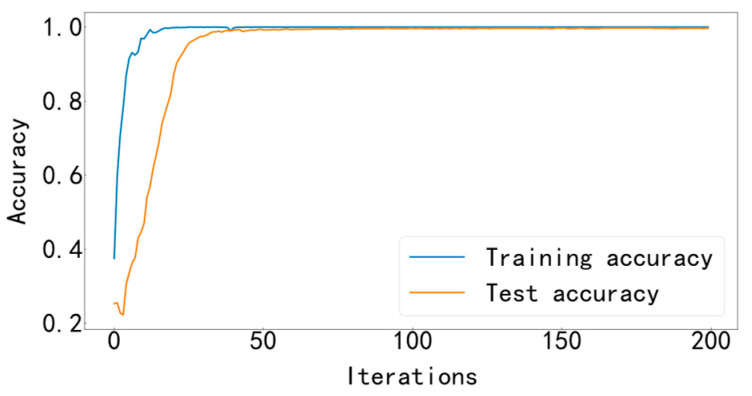
Training and testing accuracy curve.

**Figure 10 entropy-25-00737-f010:**
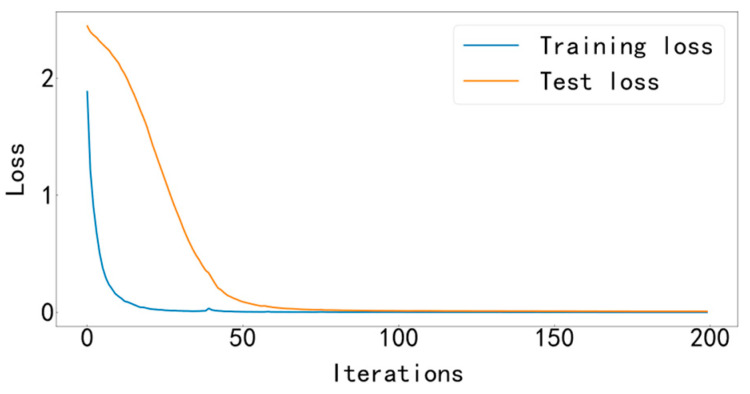
Training and testing loss curve.

**Figure 11 entropy-25-00737-f011:**
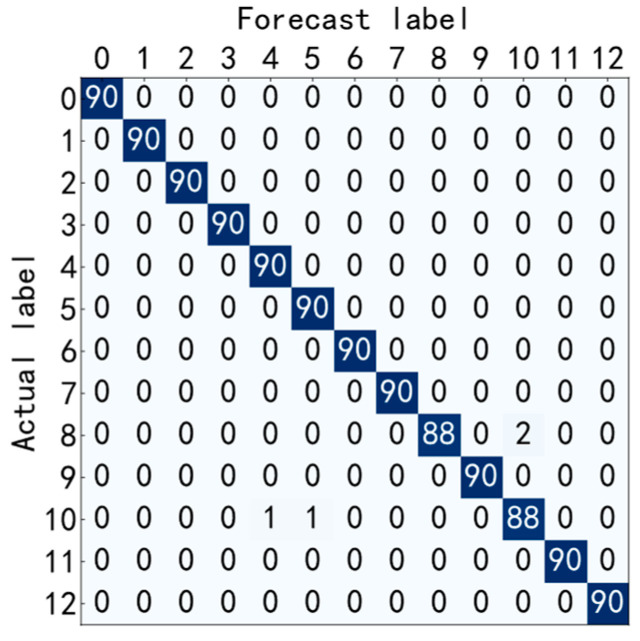
Test set confusion matrix.

**Figure 12 entropy-25-00737-f012:**
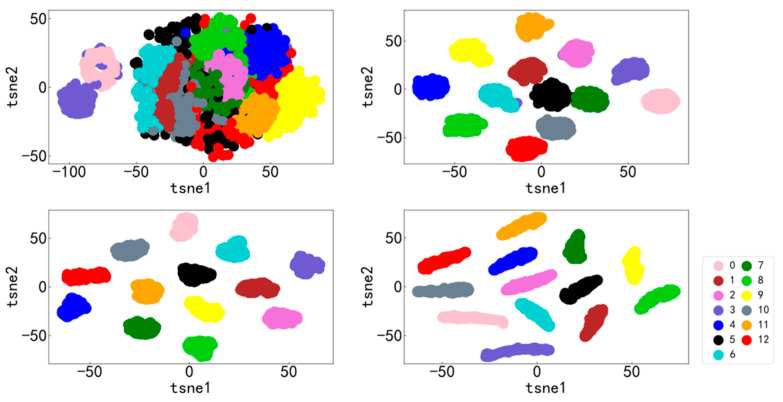
Visualization diagram of the overall training process.

**Figure 13 entropy-25-00737-f013:**
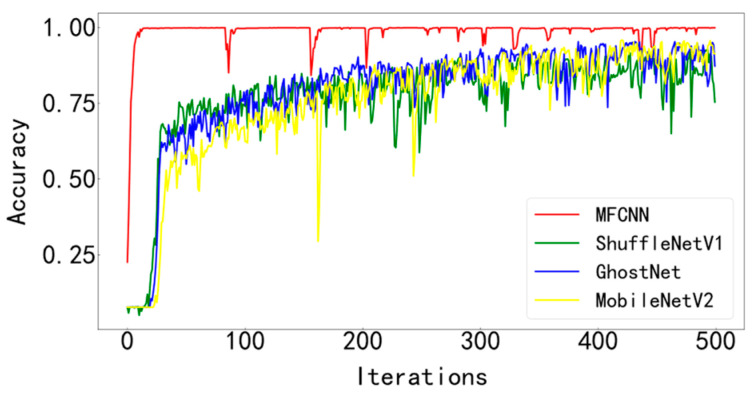
Accuracy curve of test set for various diagnostic methods.

**Figure 14 entropy-25-00737-f014:**
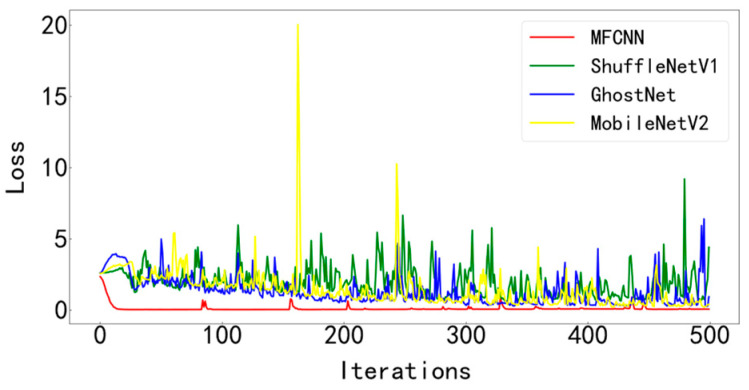
Test set loss curve of various diagnostic methods.

**Figure 15 entropy-25-00737-f015:**
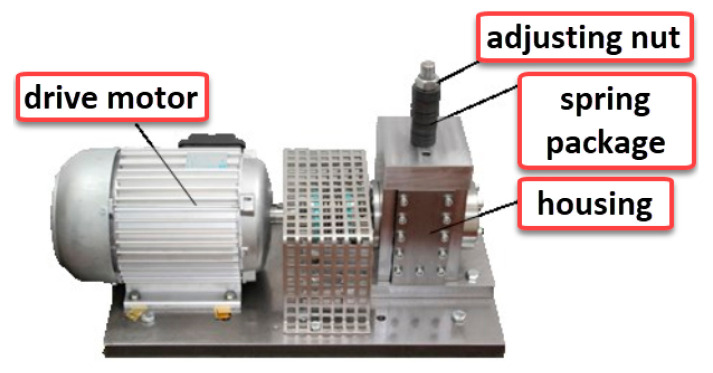
Bearing test bench.

**Figure 16 entropy-25-00737-f016:**
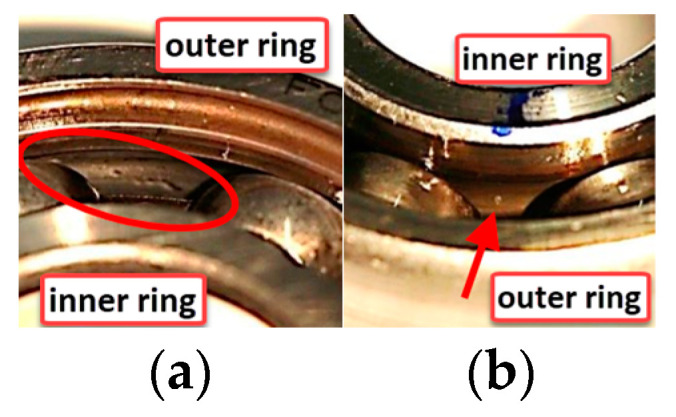
Bearing failure situation. (**a**) Outer ring fault. (**b**) Inner ring fault.

**Figure 17 entropy-25-00737-f017:**
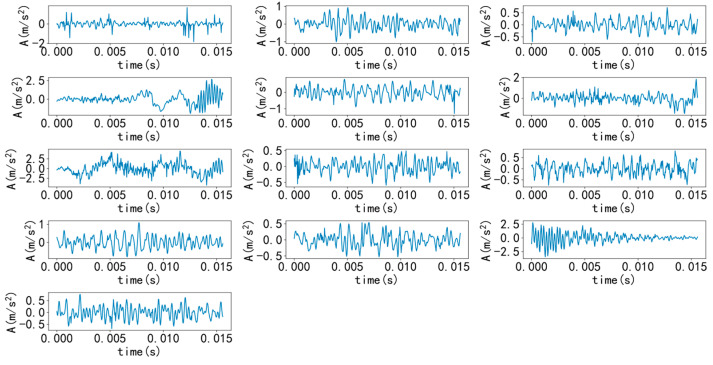
Various types of time domain diagrams.

**Figure 18 entropy-25-00737-f018:**
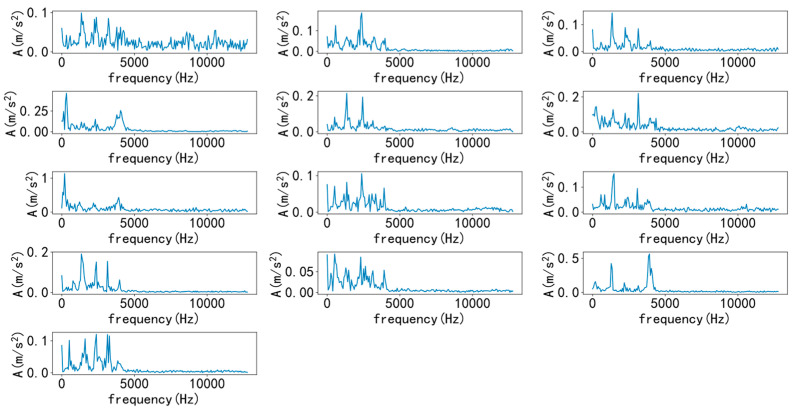
Various types of frequency domain diagrams.

**Figure 19 entropy-25-00737-f019:**
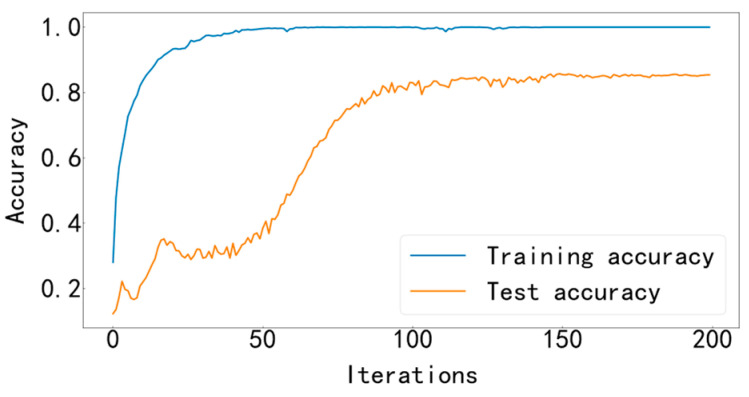
Training and testing accuracy curve.

**Figure 20 entropy-25-00737-f020:**
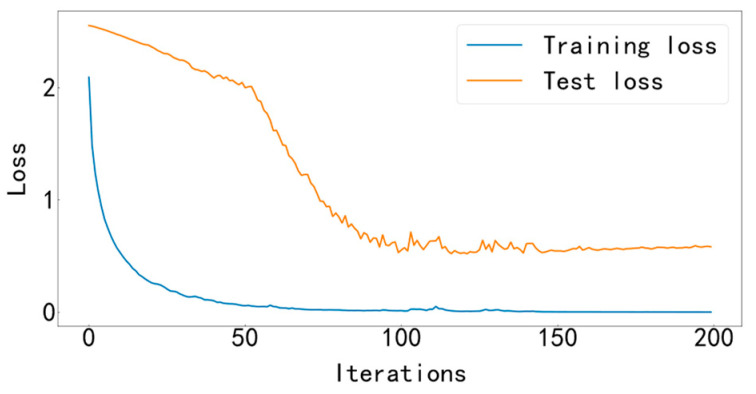
Training and testing loss curve.

**Figure 21 entropy-25-00737-f021:**
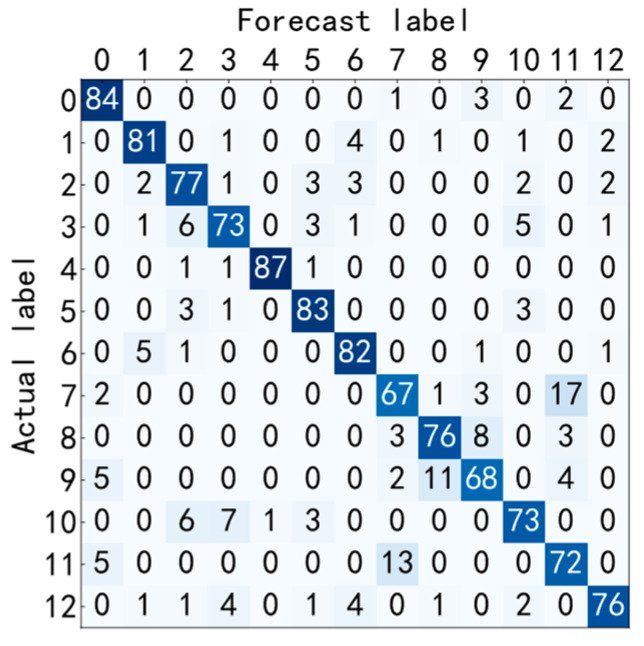
Test set confusion matrix.

**Figure 22 entropy-25-00737-f022:**
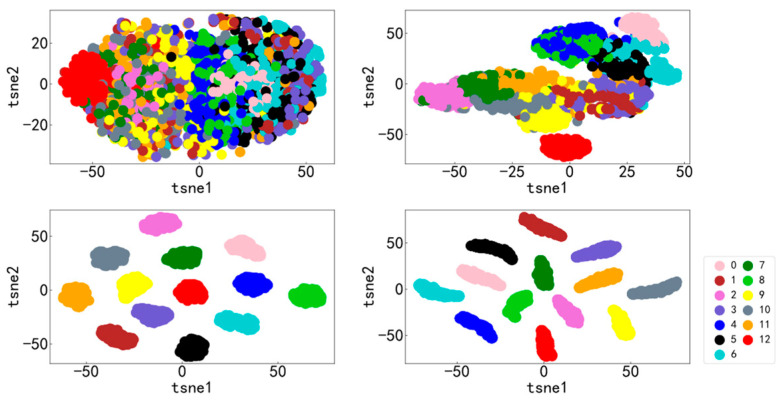
Visualization diagram of the overall training process.

**Figure 23 entropy-25-00737-f023:**
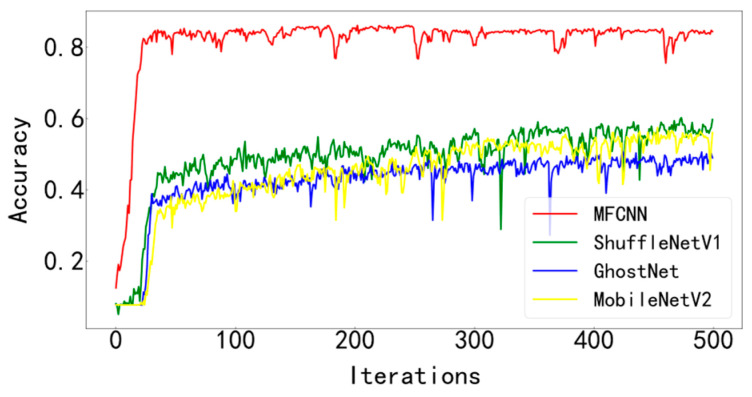
Accuracy curve of test set of various diagnostic methods.

**Figure 24 entropy-25-00737-f024:**
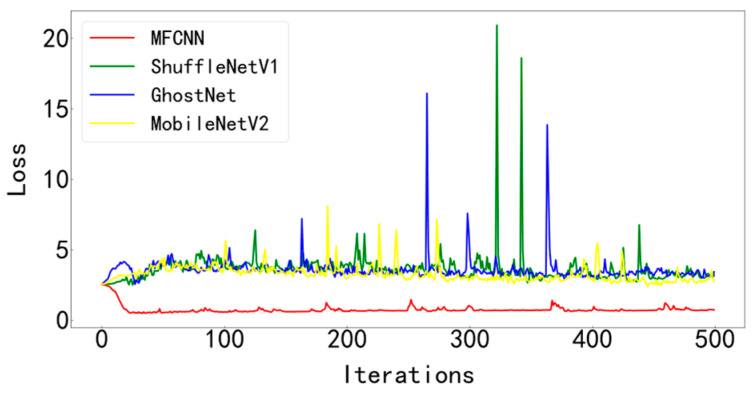
Test set loss curve of various diagnostic methods.

**Table 1 entropy-25-00737-t001:** MFCNN parameter settings.

MFCNN Network Structure	Parameter Setting
Convolution layer	16 × 16 × 16
Batch normalization layer	/
Average pooling layer	Pooled window 3 × 3, Stride 2
Convolution layer	3 × 3 × 32
Batch normalization layer	/
Average pooling layer	Pooled window 3 × 3, Stride 2
Convolution layer	3 × 3 × 64
Batch normalization layer	/
Average pooling layer	Pooled window 3 × 3, Stride 2
Global average pooling layer	/
Output layer	13 × 1

**Table 2 entropy-25-00737-t002:** LDK UER204 bearing specification sheet.

Parameter Type	Parameter Value
Inner race raceway diameter (mm)	29.30
Outer ring raceway diameter (mm)	39.80
Bearing pitch diameter (mm)	34.55
Basic dynamic load rating (N)	12,820
Ball diameter (mm)	7.92
Number of balls	8
contact angle (°)	0
Basic static load rating (KN)	6.65

**Table 3 entropy-25-00737-t003:** Data type and tag number.

Data Type	Tag Number
Normal data	0
Outer ring fault 1	1
Outer ring fault 2	2
Holder fault 1	3
Mixed fault of inner ring and outer ring	4
Inner ring fault 1	5
Outer ring fault 3	6
Holder fault 2	7
Outer ring fault 4	8
Outer ring fault 5	9
Mixed failure of inner ring, rolling element, holder, and outer ring	10
Inner ring fault 2	11
Inner ring fault 3	12

**Table 4 entropy-25-00737-t004:** Comparison of parameters of various diagnostic methods.

Diagnostic Method	Number of Parameters
MFCNN	28,429
ShuffleNetV1	952,549
GhostNet	2,605,480
MobileNetV2	2,282,981

**Table 5 entropy-25-00737-t005:** Data type and tag number.

Data Type	Tag Number
healthy bearing data	0
Outer ring damage; Pitting; Single point damage; Damage level 1	1
Outer ring damage; Plastic deform; Single point damage; Damage level 1	2
Outer ring damage; Pitting; Single point damage; Damage level 2	3
Outer ring damage; Plastic deform; Distributed damage; Damage level 1	4
Mixed damage of outer and inner rings; Pitting; Single point damage; Damage level 2	5
Mixed damage of outer and inner rings; Pitting; Distributed damage; Damage level 3	6
Mixed fault of outer ring and inner ring; Plastic deform; Distributed damage; Damage level 1	7
Inner ring damage; Pitting; Single point damage; Damage level 1	8
Inner ring damage; Pitting; Single point damage; Damage level 3	9
Inner ring damage; Pitting; Single point damage; Damage level 1	10
Inner ring damage; Pitting; Single point damage; Damage level 2	11
Inner ring damage; Pitting; Single point damage; Damage level 1	12

## Data Availability

The data used to support the findings of this study are available from the corresponding author upon request.

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
