# Peer review of "Multi-Fault Classification and Diagnosis of Rolling Bearing Based on Improved Convolution Neural Network"

_entropy, 2023, doi:10.3390/e25050737_

Round 1
Reviewer 1 Report
This paper proposes a convolution neural network for bearing multiple fault classification (MFCNN) under multiple working conditions. This method improves the traditional convolution neural network and adopts a simple structure of three-layer convolution to simplify the model. This paper is well organized and written. It can be accepted if modify the following problems.
1. The "Where" for interpretation does not require spaces, and should be "where‘’ï¼›
2. The contribution is not clear enough at least to this reviewer. It's better to give it directly in the Introduction;
3. The organizational arrangement of the paper should be given in the Introduction.
Reviewer 2 Report
In the present manuscript, the authors have proposed a Multi-fault Classification and Diagnosis of Rolling Bearing Based on Improved Convolution Neural Network. The article's overall presentation style and contents are fine; however, the authors should address and incorporate the following comments to publish it in the Journal.
1. The authors need to address the novelty and contribution of the research work in the abstract and introduction section briefly.
2. The authors need to describe the comparative advantages and innovations of the current deep learning-based approach as compared to the literature approaches. There are plenty of papers published on AI-based condition monitoring techniques.
https://doi.org/10.1007/s10845-020-01600-2 <https://doi.org/10.1007/s10845-020-01600-2>
https://doi.org/10.1016/j.isatra.2020.10.052 <https://doi.org/10.1016/j.isatra.2020.10.052>
https://doi.org/10.1016/j.neucom.2019.05.052 <https://doi.org/10.1016/j.neucom.2019.05.052>
The reviewer recommends the authors enhance the literature survey section of the current manuscript and compare the current approach's significance to the previously published studies.
3. The authors should compare the performance of the current deep learning model with the state-of-the-art models.
4. The performance of the proposed model should be validated on the public domain dataset.
5. Deep learning models are black box models. Model interpretability is a big challenge. How can it be addressed?
6. How have the authors selected the number of layers in the CNN model? How have the hyper-parameters been optimized? Which technique was used for hyper-parameters optimization?
7. What was the objective function?
8. The significance of the work is not mentioned anywhere. The authors should add the significance of this current study.
9. In the conclusion section, the limitation of the proposed methodology should be included. Also, the future scope can be added.
Reviewer 3 Report
This is an interesting piece of work. Definitely, the claim that the proposed CNN is performing better than existing CNN is remarkable but no adequate explanation neither on the math nor the physics standpoints are given why this remarkable performance.
I suggest that the authors supplement the article with the following information:
1. explains in detail that type of signals are used to train the network. Present samples of time series along with FFT and wavelet transforms so the reader can see the complexity of frequency content and whether this frequency is time dependent, thus the wavelet transform.
2. Present pictures of the faults Induced in the ball bearings.
The above will support the claim of the results.
Please comment on table 4. It seems that we have redundancy in the parameters of the other methods. Does it mean that somehow the proposed model is an optimum reduction of the other models? So does it make sense to address the issue of how to optimally reduce an initial raw CNN to an optimum one?
The article needs to address the above issues as a contribution.
Round 2
Reviewer 2 Report
Accept in present form
Reviewer 3 Report
The article is improved sufficiently for publication.